# LESS IS MORE: ADAPTIVE COVERAGE FOR SYNTHETIC TRAINING DATA

## ABSTRACT

Synthetic training data generation with Large Language Models (LLMs) offer a promising solution to the challenge of obtaining large, labeled datasets for training classifiers. When rapid model deployment is critical, such as in classifying emerging social media trends or combating new forms of online abuse tied to current events, the ability to generate training data is invaluable. While prior research has examined the comparability of synthetic data to human-labeled data, this study introduces a novel sampling algorithm, based on the maximum coverage problem, to select a representative subset from a synthetically generated dataset. Our results demonstrate that training a classifier on this contextually sampled subset achieves superior performance compared to training on the entire dataset. This "less is more" approach not only improves model accuracy but also reduces the volume of data required, leading to potentially more efficient model fine-tuning.

## 1 INTRODUCTION

In recent years, the remarkable advancement in large language models (LLMs) from companies like OpenAI (Achiam et al., 2023) or Google (Comanici et al., 2025; Team et al., 2025) have dramatically expanded the capability to generate extensive synthetic textual data. Such synthetic data promises substantial utility for training machine learning models, especially in domains where human-labeled data are prohibitively costly, inaccessible due to privacy or ethical constraints, or impractical to acquire at scale (Bunte et al., 2021; Ding et al., 2022). Consequently, synthetic data generation has quickly become an appealing alternative for tuning models for various downstream tasks, including text classification, sentiment analysis, relation extraction, and information retrieval (Meng et al., 2022).

However, the mere abundance of synthetic data does not guarantee superior model performance. Increasing evidence demonstrates that naively utilizing large synthetic datasets introduces critical pitfalls: notably, redundancy and imbalance (Gandhi et al., 2024; Liu et al., 2024; Long et al., 2024). LLM-generated samples frequently exhibit redundancy by over-representing certain common patterns or phrases, potentially saturating datasets with semantically repetitive information. Consider hate speech detection, where nuanced distinctions between offensive, sarcastic, or context-dependent language are crucial: when prompted to generate training examples, an LLM may produce many straightforwardly toxic utterances, yet underrepresent borderline, coded, or indirect forms of harm (Gandhi et al., 2024; Hao et al., 2024). Such skewed representation not only dilutes the informative value of synthetic datasets but actively harms model generalization and robustness by obscuring valuable minority cases. Consequently, models trained on these synthetic corpora risk becoming overly specialized on frequent cases, compromising predictive accuracy on more nuanced real-world scenarios.

Motivated by these gaps, we propose Adaptive Coverage Sampling (ACS), a novel method that effectively curates synthetic text datasets by carefully balancing redundancy, representational diversity, and computational efficiency. ACS uniquely frames synthetic data downsampling as a structured maximum-coverage optimization problem defined over a graph representation of the data. Specifically, synthetic text samples are first embedded into a latent semantic space, forming nodes within a complete graph where edges represent semantic similarity. Our approach leverages a binary search to systematically determine the optimal similarity threshold for edge pruning, thus inducing a sparser subgraph. Subsequently, a greedy maximum-coverage approximation algorithm selects the subset of $k$ samples maximizing representational coverage, where coverage is defined as the proportion of the dataset "covered" by the selected subset and its similarity-neighbors.

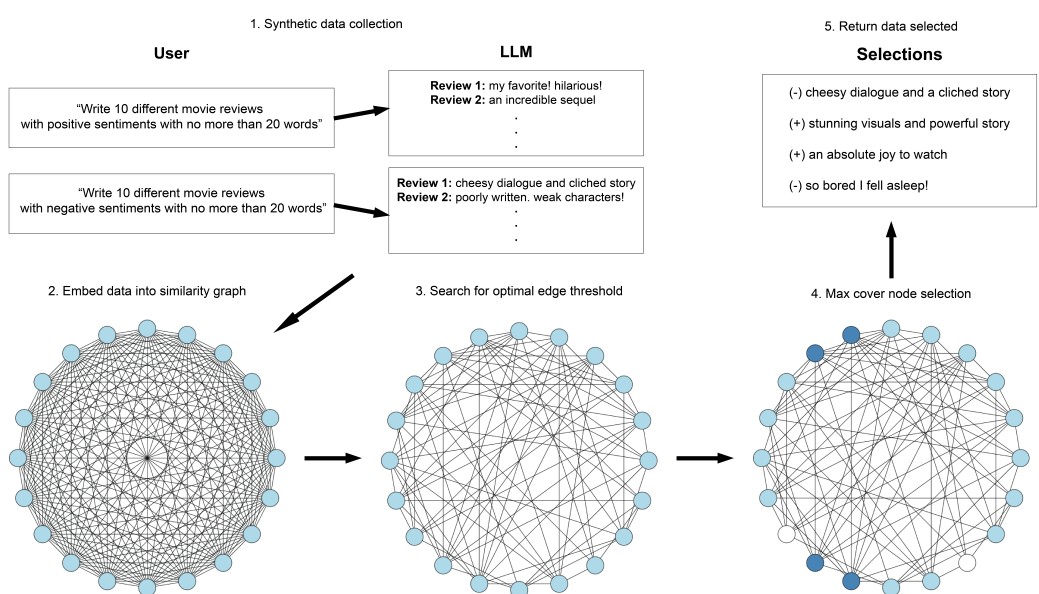

Figure 1: Overview of the ACS pipeline. (1) Prompt an LLM to generate a large pool of synthetic samples under user-specified constraints. (2) Samples are embedded into a semantic space and connected into a complete, weighted similarity graph. (3) Perform a binary search over edge-weight threshold to induce a subgraph. (4) Greedy max-cover procedure then iteratively selects the $k$ nodes (highlighted in dark blue) that together cover desired fraction of the remaining graph (uncovered nodes depicted as white). (5) Selected subset is returned for downstream model training.

A key strength of ACS lies in its use of a theoretically grounded binary search procedure to tune the pruning threshold, automating the trade-off between dataset compactness and semantic coverage. This allows the method to systematically filter out repetitive or redundant samples that might otherwise hinder model performance.

We evaluate ACS across several NLP tasks—including sentiment classification, relation extraction, and named entity recognition—and find that models fine-tuned on ACS-selected subsets match or outperform those trained on full synthetic datasets typically with just 10–30% of the original corpus. These results highlight the promise of principled data selection in synthetic data regimes: by identifying compact yet diverse training sets, ACS improves generalization while significantly reducing training compute cost. In doing so, our approach moves beyond heuristic-driven methods, offering a scalable and theoretically informed path toward more effective use of synthetic data.

## 2  RELATED WORK

**Large Language Models.** LLMs, built upon the transformer architecture introduced by (Vaswani, 2017), have transformed language processing, achieving unprecedented performance across a broad spectrum of tasks including language modeling, translation, classification, and question-answering (Brown, 2020; Rae et al., 2021; Taylor et al., 2022). These models leverage massive-scale pretraining on extensive datasets to encode rich linguistic and factual knowledge, enabling fluent and contextually relevant text generation (Team et al., 2024). Consequently, the sophistication of LLM-generated content increasingly blurs the line between synthetic and authentic human-written text (Hartvigsen et al., 2022; Sahu et al., 2022; Tang et al., 2023; Ye et al., 2022). This indistinguishability raises an intriguing question: under what conditions and to what extent can LLM-generated data replace or complement human-annotated examples for training machine learning models?

**Synthetic Training Data Generation.** High-quality datasets crucially underpin the performance and generalization capabilities of modern machine learning systems. However, acquiring diverse and representative labeled data from human annotators is frequently costly, labor intensive, and

fraught with privacy or ethical challenges (Kurakin et al., 2023; Gilardi et al., 2023; Hosking et al.; Singh et al.). Moreover, human-generated annotations inherently carry biases or inconsistencies, potentially limiting their effectiveness in certain contexts. To overcome these limitations, synthetic data generation has emerged as a promising alternative, aimed at artificially populating underrepresented data regions and mitigating biases or gaps in existing datasets (Gandhi et al., 2024; Liu et al., 2024; Li et al., 2023).

To address data scarcity in specialized or emerging domains, researchers frequently employ data augmentation techniques to enhance model robustness and accuracy (Ding et al., 2020; Wei & Zou, 2019). Moreover, semi-supervised learning (Miyato et al., 2016), multi-task learning (Glorot et al., 2011), unsupervised pretraining (Devlin, 2018; Raffel et al., 2020), and few-shot learning (Deng et al., 2020; He et al., 2021) constitute alternative frameworks for learning from limited labeled examples. However, while effective in certain contexts, these approaches typically presume access to at least some high-quality human-generated examples as seed data, limiting their broader applicability.

**Leveraging LLMs for Synthetic Data.** LLMs offer a compelling approach to synthetic data generation due to their fluency, versatility, and capacity to mimic diverse linguistic styles and content structures (Ding et al., 2022). Recent studies have demonstrated promising outcomes leveraging prompt-based methods (zero and few-shot) for generating training data for NLP tasks (Long et al., 2024). The effectiveness of synthetic datasets produced by these models depends critically on task characteristics, including the complexity of label spaces (Ding et al., 2022), the inherent subjectivity or ambiguity of the task (Li et al., 2023), and crucially, the diversity and representativeness of generated samples (Hao et al., 2024). Though the models are promising, these factors can impede naively employed models trained on synthetic datasets, potentially exacerbating redundancy and bias. Thus, underscoring the necessity of methods to carefully select or filter synthetic samples to maximize utility and minimize detrimental impacts.

**Data Filtering and Downsampling.** Filtering datasets to identify informative subsets for training constitutes a widely explored solution to the challenges posed by redundancy and imbalance, where conventional data selection techniques frequently rely on heuristic-based strategies and sample reweighting schemes (Albalak et al., 2024; Coleman et al., 2019; Kuo et al., 2024; Lin et al., 2009; Maharana et al., 2023; Paul et al., 2021; Pleiss et al., 2020; Sorscher et al., 2022; Toneva et al., 2018; Xia et al., 2022; Zheng et al.). These methods largely revolve around assigning differential importance to data points based on criteria such as correctness, informativeness, or influence on model parameters (Guo et al., 2022).

Heuristic approaches typically leverage training dynamics or statistical properties of samples. For instance, dataset cartography (Swayamdipta et al., 2020) identifies and emphasizes data points classified as difficult or ambiguous through repeated training runs. Influence functions quantify individual data sample contributions by approximating how their exclusion alters model parameters (Koh & Liang, 2017). Other methods, such as EL2N scoring (Paul et al., 2021), forgetting scores (Toneva et al., 2018), and prototypicality assessments (Sorscher et al., 2022), attempt to prioritize or prune samples based on specific diagnostic measures. Recent studies have further explored the utility of LLM-based raters to directly score or filter synthetic samples based on quality heuristics. Notably, (Chen et al., 2023) proposed AlpaGasus, demonstrating that a curated, high-quality synthetic subset significantly improves downstream model performance over full synthetic datasets. However, their approach entails repeated queries to an LLM to iteratively refine sample sets, yielding a black-box rating metric which necessitates a computational (and potentially monetary) overhead in addition to careful threshold tuning.

In contrast, our ACS methodology provides a principled, computationally efficient, and explainable solution for optimal synthetic subsets without extensive manual tuning or iterative refinement. By formulating the selection problem as a graph-based maximum coverage optimization and leveraging an adaptive binary search to systematically adjust similarity thresholds, ACS ensures theoretical rigor and practical efficacy. Crucially, ACS consistently demonstrates superior performance using significantly smaller synthetic subsets compared to prior filtering methods, thereby establishing a new benchmark for efficient and effective synthetic data utilization.

# 3 PRELIMINARIES & METHODOLOGY

In this section, we detail our comprehensive pipeline for curating a representative subset from large synthetic datasets, specifically designed to improve model training efficiency and downstream task performance. We begin by describing the generation and preprocessing of synthetic textual data, then present multiple baseline downsampling methods employed for comparative evaluation. Subsequently, we introduce and rigorously define our novel ACS method, highlighting its theoretical foundations and practical implementation. Finally, we describe our approach for fine-tuning the BERT model with the selected subset.

## 3.1 SYNTHETIC DATA GENERATION

We utilize a synthetic corpus of text generated by GPT-3.5 (Achiam et al., 2023). The corpus employed is based on established prompt templates tailored to specific downstream tasks (e.g. sentiment analysis), as detailed by prior work (Ding et al., 2022). Each dataset is balanced across labels to ensure sufficient diversity, carefully selecting an equal number of data points per label. While synthetic datasets provide vast training material, redundancy frequently arises as similar semantic content is generated repeatedly (Long et al., 2024).

## 3.2 DOWNSAMPLING METHODS.

To mitigate redundancy and maximize representational coverage, we explore several distinct downsampling techniques. Our goal is to select a subset of size $k < N$ from an initial corpus of size $N$, preserving data diversity while enhancing computational efficiency.

**Baseline Methods.** We benchmark our novel ACS approach against widely used benchmark methods. **Random** sampling henceforth refers to uniformly at random selecting $k$ samples from the corpus. **EL2N** (Paul et al., 2021) ranks samples by the average $L_2$ distance between model predictions and true labels across early training checkpoints, emphasizing persistently challenging examples. **Forgetting scores** (Toneva et al., 2018) count transitions between correct and incorrect model predictions per sample during training, emphasizing samples near the decision boundary. **Prototypicality** (Sorscher et al., 2022) which computes class-specific embeddings and prioritizes samples closest to their class centroids, capturing representative class examples. **LLM rater (AlpaGasus)** (Chen et al., 2023) employs GPT-3.5 to assign quality ratings to each synthetic input-output pair, retaining only the highest ranked samples, thereby enhancing subset quality through language-model-informed filtering. Each baseline is implemented to rank the dataset according to the respective criteria, selecting the top $k$ samples for training.

**Adaptive Coverage Sampling.** ACS introduces a graph-based max-coverage sampling technique to systematically select representative subsets. Samples are first embedded into a latent semantic space using Gecko embeddings (Lee et al., 2024), though ACS is broadly compatible with alternative embedding methods.We construct a similarity graph where each node represents a sample, and edges indicate cosine similarity exceeding a dynamic threshold. This threshold is optimized via a binary search to achieve a user-specified graph coverage level. Coverage formally quantifies representational breadth:

**Definition 3.1** (Coverage). *Let $G = (V, E)$ be a graph with vertex set $V$, edge set $E$, and self-loop for all vertices. A subset $H \subseteq V$ of size $|H| = k$ achieves coverage $c \in [0, 1]$ if*

$$\left| \bigcup_{i \in H} N_i \right| = c \cdot |V|$$

*where $N_i$ is the neighborhood of vertex $i \in H$ (ie. $i$ covers the elements of $N_i$, including itself).*

A coverage of 1.0 thus ensures every node is either selected or directly adjacent to a selected node, while lower coverage levels strategically exclude less representative samples. We leverage the following theorem guaranteeing monotonicity of an exact solution to the max cover problem with respect to similarity thresholds on the pruned graph, validating our subsequent binary search procedure.

**Theorem 3.2.** *Let $D$ be a dataset, and for each similarity threshold $s_i$, construct a similarity graph $G_i(V, E_i)$, where $V$ represents the data points and $(u, v) \in E_i$ if and only if the cosine similarity*

*between $u$ and $v$ exceeds $s_i$. Let $H_i \subseteq V$ be the set of $k$ samples selected by the max coverage algorithm on $G_i$, and let $c_i$ denote the coverage achieved by $H_i$. For any two thresholds $s_i$ and $s_j$ such that $s_j < s_i$, the similarity graph $G_j(V, E_j)$ has a coverage $c_j \geq c_i$ when maximally covered by $k$ samples.*

*Proof.* Consider two similarity thresholds $s_i$ and $s_j$ such that $s_j < s_i$. The corresponding similarity graphs $G_i(V, E_i)$ and $G_j(V, E_j)$ are constructed by adding edges between data points whose cosine similarity exceeds $s_i$ and $s_j$, respectively. Since $s_j < s_i$, it follows that $E_i \subseteq E_j$; that is, $G_j$ includes all the edges from $G_i$, possibly with additional edges.

Now, let $H_i \subseteq V$ be the set of $k$ samples selected by the max coverage algorithm on $G_i$, which achieves coverage $c_i$. The coverage $c_i$ is defined as the proportion of vertices in $V$ that are adjacent to at least one vertex in $H_i$. Since $E_i \subseteq E_j$, the set of neighbors of each vertex in $H_i$ in $G_i$ is a subset of the neighbors of the same vertex in $G_j$. Therefore, the coverage achieved by $H_i$ in $G_j$ is at least as large as the coverage in $G_i$. More formally, if $H_j$ is the set of $k$ samples selected by the max coverage algorithm on $G_j$, we have:

$$c_j = \left| \bigcup_{v \in H_j} N_j(v) \right| \quad \text{and} \quad c_i = \left| \bigcup_{v \in H_i} N_i(v) \right|,$$

where $N_j(v)$ and $N_i(v)$ denote the neighborhoods of $v$ in $G_j$ and $G_i$, respectively. Since $E_i \subseteq E_j$, we have $N_i(v) \subseteq N_j(v)$ for all $v \in V$, implying that the coverage in $G_j$ is at least as large as the coverage in $G_i$. Therefore, $c_j \geq c_i$. □

The monotonicity of coverage allows us to find the largest similarity threshold that achieves a coverage equal to, or greater than, the target coverage. This thresholding ensures that the max coverage component of ACS focuses on the most relevant and diverse samples to achieve the target coverage. We note that the max coverage problem is NP-hard (Feige, 1998), and that our implementation uses the greedy approximation (Hochbaum, 1996) which is not guaranteed to be monotonic. However, we show that, in practice, this monotonicity persists (see Section 4.1).

Leveraging this result, we conduct a binary search on the similarity threshold for edge pruning and execute the greedy max cover algorithm Biabani et al. (2023); Lin et al. (2009). Specifically, we sequentially select the node of highest degree, add the selected node and all of its neighbors to the set of "covered nodes" and repeat until $k$ nodes are selected. We subsequently compute the coverage of the full dataset from the selected subset and, based on this coverage's deviation from the target, adjust the threshold in accordance with the binary search until convergence. The $k$ selected points from the max cover execution on the optimally pruned graph are finally returned.

To ensure scalability and enhance representational diversity, we impose a maximum nearest neighbors constraint per node, significantly reducing computational complexity and ensuring effective coverage. Specifically, we define a strict constraint $d_{\max}$, a bound ensuring sufficient but limited graph connectivity, derived via the extended pigeonhole principle: $d_{\max} > cN/k$. This constraint further improves computational tractability and sample diversification, analgous to scalability techniques like Locality-Sensitive Hashing (LSH) with limited bucket sizes (Chen et al., 2022; Shekkizhar et al., 2023).

### 3.3 COMPARATIVE EXPERIMENTS

After generating and downsampling the synthetic dataset to obtain $k$ training samples, we employ two comparative measures. First, we fine-tune a BERT model (Devlin, 2018) on the selected subset and report the F1-scores as a function of the number of subsamples selected for model training[1]. We use the $\text{BERT}_{\text{base}}$, uncased model (108 million parameters) and fine-tune it for multiple epochs (the exact number is defined for each respective experiment in Section 5). The model's weights are mostly initialized using pre-trained weights, while the parameters of the final classification layer (2048 units) are randomly initialized. Specifically, the weights of this layer are initialized from a normal distribution with a mean of 0 and a standard deviation of 0.02, following standard practices

---

[1]We here use F1, rather than accuracy, to remain robust to potential class imbalances in the test sets.

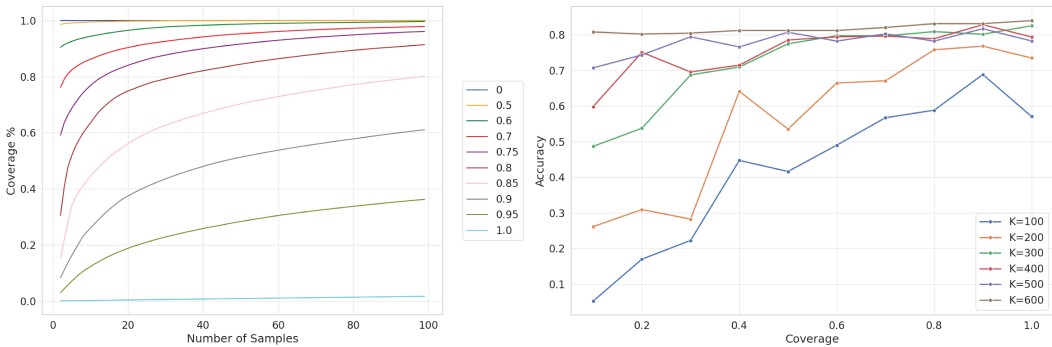

Figure 2: (L) Coverage of data increases with $k$ or when decreasing the similarity threshold. Colors correspond to the fixed similarity thresholds depicted in the legend. (R) Model accuracy as a function of coverage level for the sentiment analysis tasks. Performance peaks at a coverage level below 1.0.

for fine-tuning transformer-based models.(Devlin, 2018; Dodge et al., 2020; Lan, 2019; Liu, 2019). The fine-tuning process uses a batch size of 16, a learning rate of $2 \times 10^{-5}$, and a dropout rate of 0.1. All experiments are conducted on a high-performance GPU cluster with 16GB of RAM, with $n = 5$ distinct random seeds used for model initialization. Details of the implementation, including all hyperparameters, are provided in the supplementary material, along with the training codes.

Second, we compute the self-bilingual evaluation understudy (or SelfBLEU) metric as a quantifiable measure of subset diversity (Zhu et al., 2018). Thismetric computes word similarity between sentences or documents within a dataset. A higher SelfBLEU score indicates a dataset with higher self-similarity, thus the *reciprocal* is used as a diversity measure.

## 4 EMPIRICAL ANALYSIS OF ACS

In this section, we empirically validate critical aspects of our sampling method. Specifically, we first verify the empirical monotonicity of coverage as a function of similarity threshold for the greedy approximate algorithm for the max coverage problem, aligning with the theoretical guarantees provided by Theorem 3.2. We then systematically identify and analyze the coverage parameter value, demonstrating that coverage below 1.0 consistently yields better performance in downstream tasks.

### 4.1 EMPIRICAL VALIDATION OF MONOTONICITY

A central assumption is ACS is that coverage monotonically increases or remains constant as the similarity threshold decreases, as formally established for the exact max coverage solution. To confirm this assumption's practical validity under the greedy approximation algorithm (Hochbaum, 1996), we conducted detailed empirical experiments across varying similarity thresholds.

We focus initially on the synthetic textual data generated to emulate the SST2 sentiment analysis task (Socher et al., 2013). This synthetic dataset comprises short movie reviews labeled as positive or negative sentiments. Additional validation on other datasets, is provided in the supplementary materials.Each text sample was first embedded into a latent semantic space using Gecko embeddings (Lee et al., 2024). Subsequently, similarity graphs were constructed for multiple fixed similarity thresholds, after which the greedy max-coverage approximation algorithm was executed to select subsets of varying sizes $k$. As illustrated in the left-hand plot of Figure 2, coverage consistently exhibits monotonic behavior: as the similarity threshold decreases (adding more edges), coverage either remains constant or strictly increases, validating our core theoretical assumption in practical scenarios. Notably, the maximum possible coverage (full coverage, $c = 1$) is achieved quickly at lower thresholds, while all plots achieve a minimal coverage of $c = k/N$.

## 4.2 Determining the Optimal Coverage Level

While full coverage ($c = 1$) intuitively seems optimal, in practice, we demonstrate that lower coverage values yield better model performance. We hypothesize that this is due to the exclusion of redundant or noisy samples. To systematically investigate this, we varied the coverage parameter across a broad range of values, maintaining a fixed subset size of $k$ for the synthetic SST2 dataset (with analogous findings for additional tasks reported in the Appendix A.1).

For each coverage setting, ACS selected a subset of exactly $k$ samples which achieved an effective coverage of the target. Using these subsets, we fine-tuned BERT$_{\text{base}}$ models and evaluated their accuracy on a human-annotated test set. The resulting accuracy trends are presented on the right Figure 2. Notably, accuracy significantly improves as the target coverage increases from lower levels, reflecting greater representational completeness. However, accuracy consistently peaks before reaching full coverage, with performance slightly deteriorating at or near the full coverage ($c = 1$). These results robustly support our assertion that carefully selecting subsets with moderate coverage offers superior model generalization and training efficiency.

## 5 Fine-Tuning for Downstream Tasks

In this section, we rigorously evalute the performance of ACS against several established baseline downsampling methods on multiple NLP benchmarks. Specifically, we assess sequence-level tasks (sentiment analysis and relation extraction) and a token-level task (named entity recognition), demonstrating ACS's consistent advantages in terms of model performance and diversity of selected subsets. These evaluations compliment one another: improved performance corresponding to higher diversity in the selected subsets and vice versa. We note that, while our analysis explores performance across a range of subset sizes ($k$), in a practical application, the optimal value of k could be determined efficiently by monitoring model performance on a held-out validation set.

### 5.1 Sequence-Level Tasks

**Sentiment Analysis.** We first evaluate our approach on the binary sentiment classification task using the synthetic corpus from (Ding et al., 2022), designed to emulate the SST2 dataset (Socher et al., 2013). This dataset contains $N = 6,000$ synthetic movie reviews, equally split between positive and negative sentiments. Following the prior literature (Ding et al., 2022), we fine-tune a BERT$_{\text{base}}$ model for 32 epochs (with early stopping) on subsets selected by each downsampling method.

Figure 3 compares the performance (F1-score) of ACS against the baseline methods, averaging results over five random initializations of the BERT classification layer weights. ACS consistently outperforms alternative methods across all subset sizes, with particularly notable improvements at smaller subset sizes. Remarkably, ACS achieves performance comparable to training on the *full* synthetic dataset (black dashed line) **using only approximately 10% of the data**, underscoring its effectiveness in isolating highly informative samples. We note that while ACS performs the best, all methods (apart from random) yield aggressively pruned datasets which can match performance on the full dataset. This suggests that for the simpler task of positive or negative sentiment detection, only a few meaningful examples are needed to train a sophisticated classifier to effectively categorize the inputs. To further elucidate why ACS in particular achieves superior performance, Figure 3 further plots diversity (inverse SelfBLEU score) across subset sizes. ACS-selected subsets consistently exhibit greater diversity compared to baseline methods, strongly correlating with improved downstream task performance. This enhanced diversity seems to mitigate redundancy and better equips models to generalize effectively, particularly when training data sizes are limited.

**Relation Extraction.** Relation extraction, exemplified by the FewRel dataset (Han et al., 2018), represents a significantly more challenging classification task due to its large set of 64 distinct relation labels. The task involves predicting the labeled relation between two marked entities within a sentence, necessitating both greater diversity and precision in the synthetic data generation process. For instance, the sentence, "Chester Alan Arthur, 21st President of the United States, died of this disease on November 18, 1886," could be labeled with the relation "head of government" to capture the connection between Arthur and his role as President. This increased complexity necessitates careful selection of diverse and informative examples. We employ the synthetic corpus of relation extraction data from (Ding et al., 2022), uniformly sampling $N = 12,800$ examples spread across

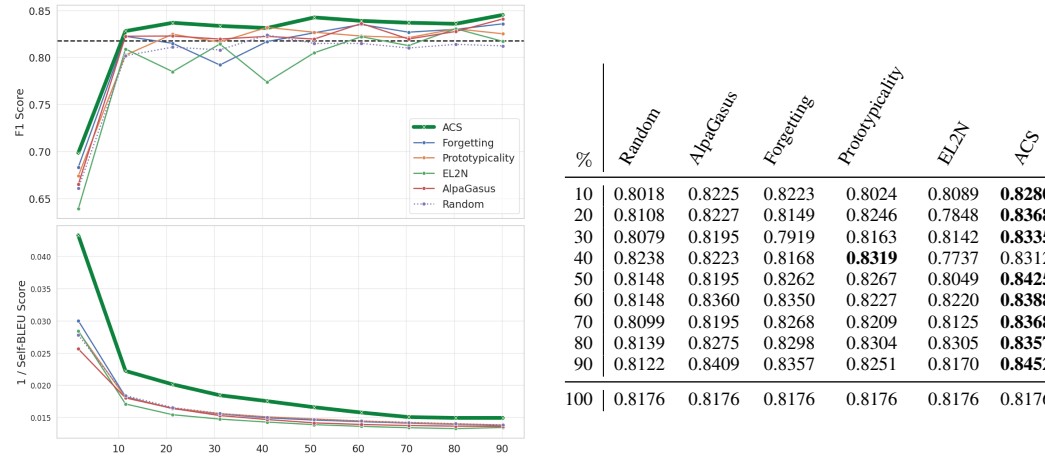

Figure 3: (L) F1 scores (top) and SelfBLEU diversity (bottom) for SST2 as a function of subset size, comparing downsampling methods. Horizontal dotted line represents model performance when trained on all available data (no pruning). (R) Tabulated F1 results corresponding with plots.

| % | Random | AlpaGasus | Forgetting | Prototypicality | EL2N | ACS |
|---|---|---|---|---|---|---|
| 10 | 0.8018 | 0.8225 | 0.8223 | 0.8024 | 0.8089 | **0.8280** |
| 20 | 0.8108 | 0.8227 | 0.8149 | 0.8246 | 0.7848 | **0.8368** |
| 30 | 0.8079 | 0.8195 | 0.7919 | 0.8163 | 0.8142 | **0.8335** |
| 40 | 0.8238 | 0.8223 | 0.8168 | **0.8319** | 0.7737 | 0.8312 |
| 50 | 0.8148 | 0.8195 | 0.8262 | 0.8267 | 0.8049 | **0.8425** |
| 60 | 0.8148 | 0.8360 | 0.8350 | 0.8227 | 0.8220 | **0.8388** |
| 70 | 0.8099 | 0.8195 | 0.8268 | 0.8209 | 0.8125 | **0.8368** |
| 80 | 0.8139 | 0.8275 | 0.8298 | 0.8304 | 0.8305 | **0.8357** |
| 90 | 0.8122 | 0.8409 | 0.8357 | 0.8251 | 0.8170 | **0.8452** |
| 100 | 0.8176 | 0.8176 | 0.8176 | 0.8176 | 0.8176 | 0.8176 |

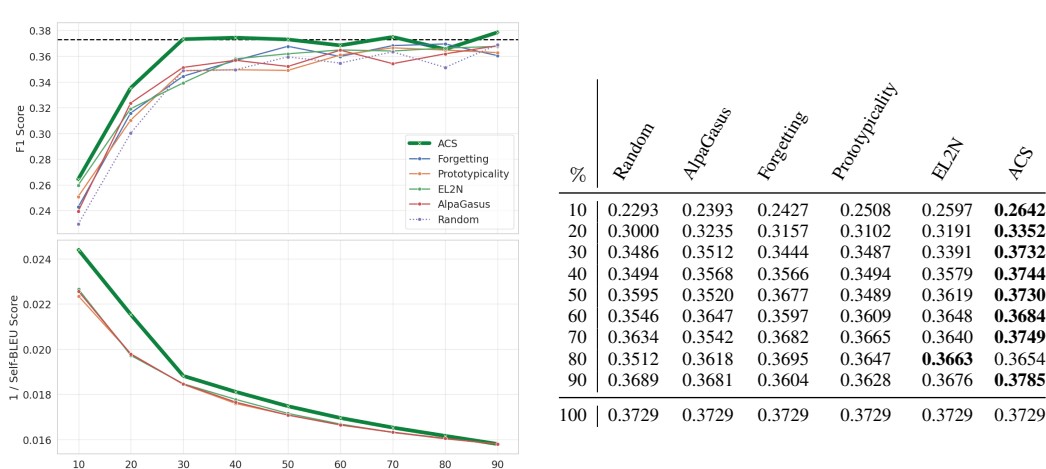

Figure 4: (L) F1 scores (top) and SelfBLEU diversity (bottom) for FewRel as a function of subset size, comparing downsampling methods. Horizontal dotted line represents model performance when trained on all available data (no pruning). (R) Tabulated F1 results corresponding with plots.

| % | Random | AlpaGasus | Forgetting | Prototypicality | EL2N | ACS |
|---|---|---|---|---|---|---|
| 10 | 0.2293 | 0.2393 | 0.2427 | 0.2508 | 0.2597 | **0.2642** |
| 20 | 0.3000 | 0.3235 | 0.3157 | 0.3102 | 0.3191 | **0.3352** |
| 30 | 0.3486 | 0.3512 | 0.3444 | 0.3487 | 0.3391 | **0.3732** |
| 40 | 0.3494 | 0.3568 | 0.3566 | 0.3494 | 0.3579 | **0.3744** |
| 50 | 0.3595 | 0.3520 | 0.3677 | 0.3489 | 0.3619 | **0.3730** |
| 60 | 0.3546 | 0.3647 | 0.3597 | 0.3609 | 0.3648 | **0.3684** |
| 70 | 0.3634 | 0.3542 | 0.3682 | 0.3665 | 0.3640 | **0.3749** |
| 80 | 0.3512 | 0.3618 | 0.3695 | 0.3647 | **0.3663** | 0.3654 |
| 90 | 0.3689 | 0.3681 | 0.3604 | 0.3628 | 0.3676 | **0.3785** |
| 100 | 0.3729 | 0.3729 | 0.3729 | 0.3729 | 0.3729 | 0.3729 |

all relation labels in accordance with the FewRel dataset. The BERT$_{base}$ model is fine-tuned over 3 epochs as in the prior work.

Figure 4 presents the F1-score results on the synthetic FewRel dataset, clearly demonstrating that ACS consistently surpasses the baseline methods at nearly all data subsampling proportions. Similar to the sentiment analysis task, ACS achieves competitive or superior performance using just 30% of the available synthetic data. Figure 4 provides additional support by showing that subsets selected by ACS, again, obtain substantially lower SelfBLEU scores, indicating greater representational diversity. This enhanced diversity is particularly valuable for relation extraction, which benefits from nuance and varied training examples to better capture the complex semantic relations between entities.

## 5.2 TOKEN-LEVEL TASK: NAMED ENTITY RECOGNITION

We lastly validate ACS on the token-level named entity recognition (NER) task using a synthetic corpus generated to match the AI domain split of CrossNER (Liu et al., 2021). This task involves

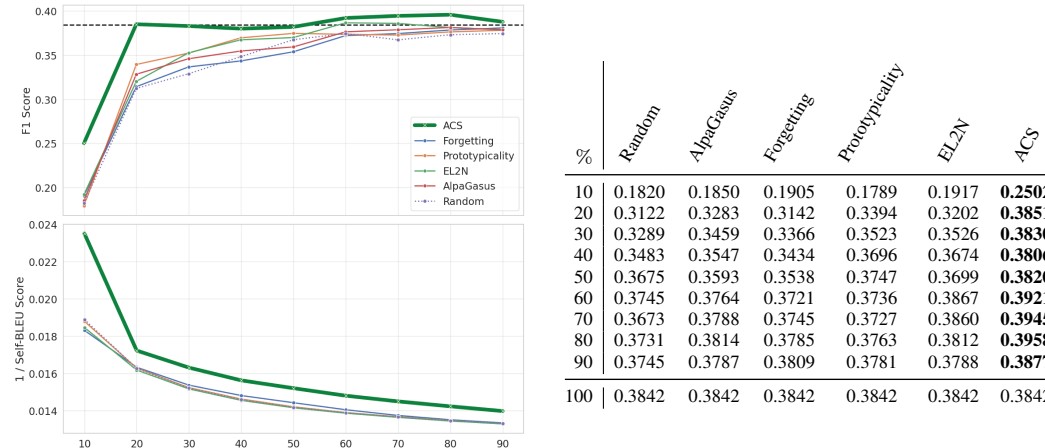

Figure 5: (L) F1 scores (top) and SelfBLEU diversity (bottom) for CrossNER as a function of subset size, comparing downsampling methods. Horizontal dotted line represents model performance when trained on all available data (no pruning). (R) Tabulated F1 results corresponding with plots.

labeling each token in a sentence with one of 14 distinct entity classes or a null identifier. For example, on the sentence: "We evaluated BERT using the SQuAD benchmark and compared its performance with BiDAF on multiple F1-score metrics." a classifier would have to mark the relevant tokens (BERT, SQuAD, BiDaF, F1-score) with the labels (Tool, Dataset, Tool, Metric) respectively. The synthetic corpus used here contains $N = 3,000$ sentences, each carefully generated to reflect diverse entity mentions. We crucially highlight for this *token-level* classification task, we still deploy ACS on the *sentence* embeddings to isolate the most representative samples. The selected sentences are subsequently parsed back into their tokenization for classification. We fine-tune a BERT$_{base}$ model specific to the NER task (Rajapakse et al., 2024) over 50 epochs, applying early stopping to prevent overfitting.

Figure 5 illustrates ACS's performance on the token-level classification task. Using only 20% of the original synthetic dataset, ACS achieves accuracy comparable to training on the entire dataset. Furthermore, ACS consistently selects subsets with notably greater diversity, as evidenced by lower SelfBLEU scores compared to baselines. This confirms ACS's capability to effectively capture a wide representation of the dataset, even for precise token-level predictions.

## 6 DISCUSSION

Our experiments convincingly demonstrate that ACS effectively distills large synthetic datasets into smaller, highly representative subsets, significantly enhancing model training efficiency and accuracy. Several distinctive strength set ACS apart from existing downsampling and filtering methods. First, ACS reliably identifies remarkably small subsets—often around 20% or even less of the original synthetic dataset—that allow models to achieve performance matching or surpassing that of models trained on the full dataset. This capability underscores the potential efficiency gains and practical utility of ACS in real-world scenarios, especially when computational resources or training time are limited. Second, unlike many alternative methods that require fitting multiple models or extensive hyperparameter tuning to gauge sample importance, ACS does not depend on repeated training iterations. Instead, our method leverages a straightforward binary search on a similarity graph. Third, ACS does not rely on label information during the subset selection phase, making it broadly applicable to both supervised and unsupervised scenarios. This feature notably enhances its versatility, enabling effective deployment in diverse data scenarios without requiring preliminary labeling efforts. Lastly, ACS explicitly focuses on identifying optimal *collections* of data points rather than individual samples with maximal individual contribution. This collection oriented approach ensures that the selected subsets comprehensively represent the overall dataset diversity and structure, rather than emphasizing potentially redundant or outlier points that individually maximize some criterion. As such, ACS offers a robust, efficient, and versatile approach to synthetic data

distillation, delivering substantial improvements in downstream task performance through highly informative and diverse subsets.

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

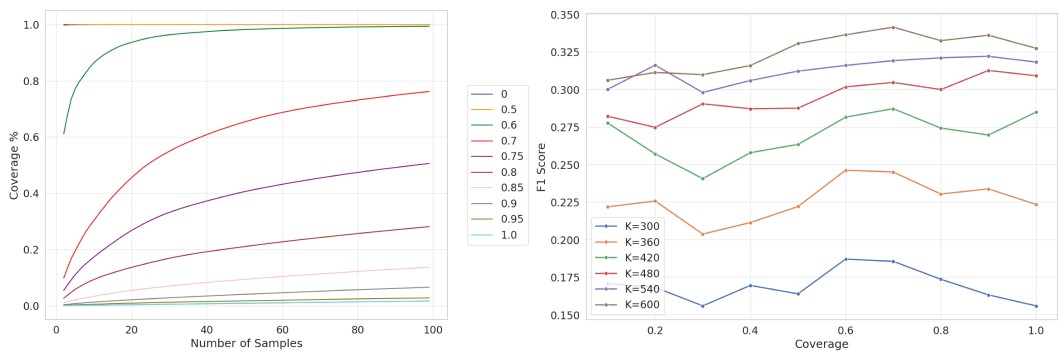

Figure 6: Empirical results for the FewRel dataset. (L) Coverage of data increases with $k$ or when decreasing the similarity threshold. Colors correspond to the fixed similarity thresholds depicted in the legend. (R) Model accuracy as a function of coverage level for the sentiment analysis tasks. Performance peaks at a coverage level below 1.0.

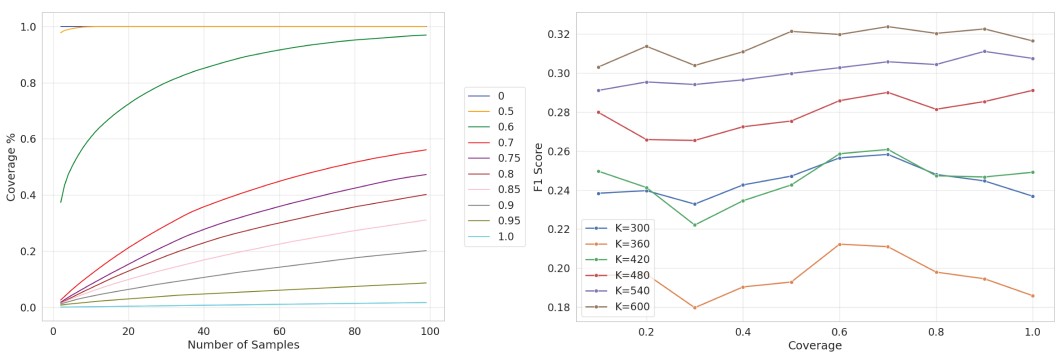

Figure 7: Empirical Results for the CrossNER dataset. (L) Coverage of data increases with $k$ or when decreasing the similarity threshold. Colors correspond to the fixed similarity thresholds depicted in the legend. (R) Model accuracy as a function of coverage level for the sentiment analysis tasks. Performance peaks at a coverage level below 1.0.

## A    OMITTED RESULTS

We here present the empirical analysis of Section 4 on the FewRel and CrossNER datasets. We further present a sensitivity analysis to the max degree parameter for all of the datasets.

### A.1    EMPIRICAL ANALYSIS OF ACS

We begin with the empirical ACS validation of Section 4 for the remaining datasests. In both instances, we observe consistent monotonicity in the coverage as a function of $k$-selection with decreasing similarity thresholds, as well as improved downstream task performance with coverage values less than 1.0. Figure 6 presents the empirical results for the FewRel dataset and Figure 7 for CrossNER. In both instances, the greedy approximation to max coverage exhibits monotonicity as needed for the binary search procedure. We further see that full coverage is non-optimal in most instances, further motivating our usages of coverage = 0.9 throughout the experimental results.

## B    SCALABILITY OF ADAPTIVE COVERAGE SAMPLING

In large-scale settings, the computational cost of optimizing the similarity threshold $\tau$ for ACS can become prohibitive due to the $O(n^2)$ complexity of evaluating pairwise similarities. Though we can speed up such computations with methods such as Locality Sensitive Hasing (LHS) or hop-spanner methods (Carey et al., 2022; Epasto et al., 2021; Halcrow et al., 2020), we further propose a scalable

variant of ACS that conducts threshold selection on a small random subset of the data. For a desired downsampling value of $k \ll N$, we uniformly at random select a small subgraph of $N' < N$ nodes and run the ACS procedure on the reduced instance. Once the optimal edge similarity threshold $\tau^*$ is identified on this subset, it is reused to construct the similarity graph and perform ACS on the *full* dataset. This approach significantly reduces computational cost while maintaining effective coverage.

Formally, let $G = (V, E)$ be the similarity graph constructed on the full dataset, where edges are defined between points with similarity exceeding a threshold $\tau$. Let $V' \subset V$ denote a uniformly random subsample of size $N'$, and let $G' = (V', E')$ be the induced subgraph. For any subset $S \subset V$, we define the normalized coverage as the fraction of nodes in $V$ that are neighbors of some node in $S$ under threshold $\tau$. We proceed to show that threshold tuning on the subsample generalizes well to the full dataset, in the following proposition.

**Proposition B.1.** *Let $S_0$ be a fixed subset of nodes. Let $U_0 = \{u \in V | \exists v \in S_0 : sim(u, v) \geq \tau\}$ be the set of all nodes in $V$ covered by $S_0$. The true coverage of $S_0$ is $C(S_0; V) = \frac{|U_0|}{|V|}$. Let $V' \subset V$ be a uniform random sample of $N'$ nodes drawn without replacement. The sample coverage is $C(U_0 \cap V'; V) = \frac{|U_0 \cap V'|}{|V'|}$. Then, with probability at least $1 - \delta$:*

$$|C(S_0; V) - C(U_0 \cap V'; V')| \leq \varepsilon$$

*where $\varepsilon = \sqrt{\frac{\ln(2/\delta)}{2N'}}$.*

*Proof.* We have a population of size $|V| = N$ containing a sub-population of "covered" nodes of size $|U_0|$. The true proportion of covered nodes is $p = \frac{|U_0|}{N} = C(S_0; V)$. We draw a random sample $V'$ of size $N'$ without replacement and calculate the sample proportion of covered nodes, which is $\hat{p} = C(U_0 \cap V'; V')$. From here, we can apply Hoeffding's inequality for sampling without replacement to bound the deviation of the sample proportion from the true value:

$$\mathbf{Pr}\left[|\hat{p} - p| \geq \varepsilon\right] \leq 2\exp(-2N'\varepsilon^2).$$

To ensure that the probability of the error exceed $\varepsilon$ is no more than a failure probability of $\delta$, we equate the right hand side of the above equation to $\delta$ and solve for $\varepsilon$ to obtain the desired result. $\square$

This proposition establishes that threshold selection on a small sample yields an accurate coverage estimate on the full dataset, with the accuracy improving for larger datasets.

To validate this claim empirically, we conducted a series of experiments across the datasets used in the main text (sentiment analysis, relation extraction, and named entity recognition). In each setting, we selected a random subset of the data at varying proportions, ranging from very small to nearly the full dataset. For each subset, we used binary search to identify the threshold $\tau^*$ such that the greedy ACS procedure on the subset achieved a fixed target of 90% coverage with $k$ examples. We then applied this same threshold $\tau^*$ to construct the similarity graph for the full dataset and ran the greedy max coverage to select a size-$K$ subset, measuring the resulting coverage over all data points.

Figure 10 summarizes the results. Each plot corresponds to a different dataset (SST2, FewRel, or CrossNER). The x-axis represents the fraction of the dataset used to compute the optimal threshold, and the y-axis shows the actual coverage obtained on the full dataset using that threshold. A shaded band indicates an $\varepsilon$-envelope centered at the target coverage of 90% where $\varepsilon$ is set to be $5 \times 10^{-3}$. Across all settings, we observe that even small subsamples, often less than 20% of the full dataset, yield thresholds that generalize well. As the sample size increases, the coverage rapidly converges to the target, and variance remains low throughout.

These results provide strong empirical support for the scalable ACS approach. By selecting a threshold on a small, randomly drawn subset, we can achieve nearly identical coverage behavior on the full dataset, enabling efficient and accurate training data selection in large-scale scenarios without repeated expensive graph construction or threshold tuning.

We note that the above experiments, in line with Proposition B.1 do not impose any max degree constraints on the similarity graph. We demonstrate that even when such constraints are imposed, the scalability of optimal threshold remains. In Figure 10, we again impose the max degree constraint of $2 \cdot c \cdot N / k$ and set a target coverage of 0.5.

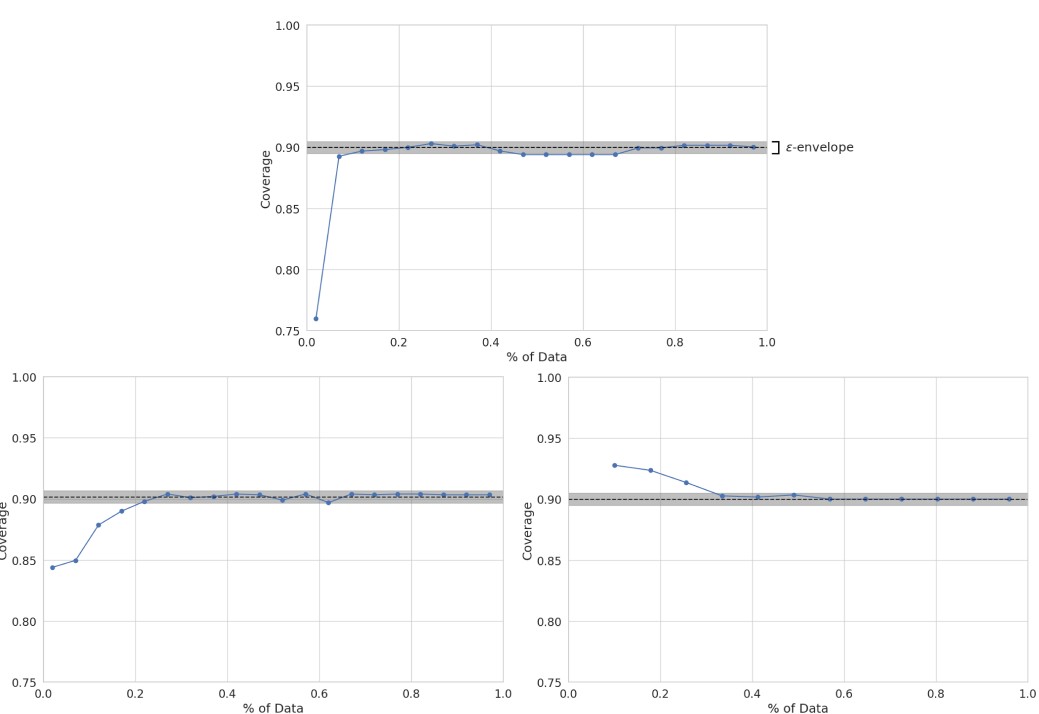

Figure 8: Coverage transfer from subsample to full dataset. Each point corresponds to a threshold $\tau^*$ optimized on a random subset of a given size and evaluated for coverage on the full dataset. The gray band denotes a small tolerance range around the 90% target. Results show threshold transfer achieves accurate and stable coverage across various dataset sizes. (Top Left) SST2, (Bottom Left), CrossNER, (Bottom Right) FewRel.

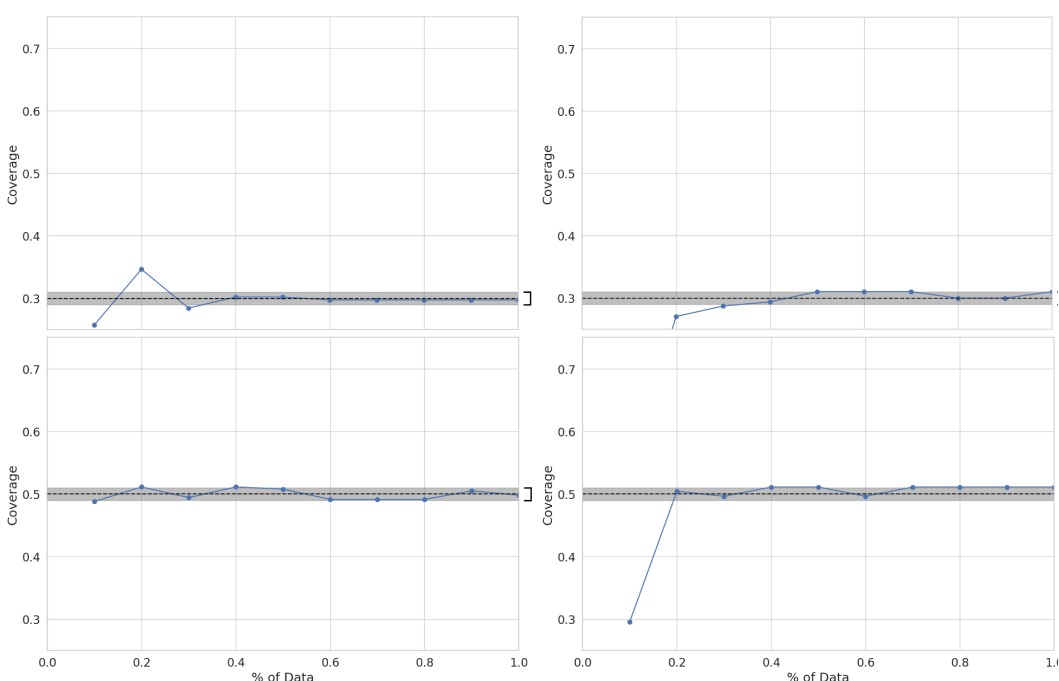

Figure 9: Coverage transfer from subsample to full dataset. Each point corresponds to a threshold $\tau^*$ optimized on a random subset of a given size and evaluated for coverage on the full dataset. The gray band denotes a small tolerance range around the $30\%$ and $50\%$ targets. Results show threshold transfer achieves accurate and stable coverage across various dataset sizes. (Left) SST2, and (Right) CrossNER.

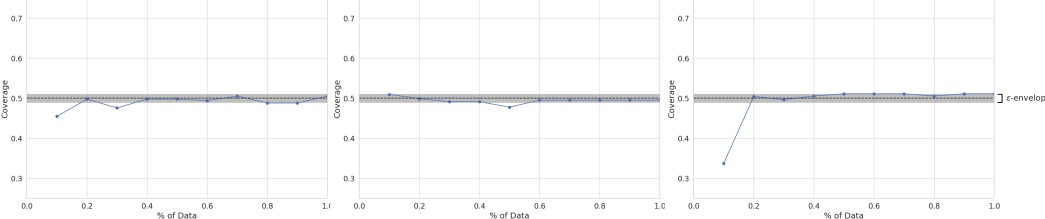

Figure 10: Coverage transfer from subsample to full dataset. Each point corresponds to a threshold $\tau^*$ optimized on a random subset with max degree constraint of a given size and evaluated for coverage on the full dataset. The gray band denotes a small tolerance range around the $50\%$ target. Results show threshold transfer achieves accurate and stable coverage across various dataset sizes. (Left) SST2, (Middle) FewRel and (Right) CrossNER.

| % | Random | AlpaGasus | Forgetting | Prototypicality | EL2N | $k$-Means | SemDeDup | ACS |
|---|--------|-----------|------------|-----------------|------|-----------|----------|-----|
| 10 | 0.8018 | 0.8225 | 0.8223 | 0.8024 | 0.8089 | 0.8028 | 0.8140 | **0.8280** |
| 20 | 0.8108 | 0.8227 | 0.8149 | 0.8246 | 0.7848 | 0.8175 | 0.8201 | **0.8368** |
| 30 | 0.8079 | 0.8195 | 0.7919 | 0.8163 | 0.8142 | 0.8198 | 0.8274 | **0.8335** |
| 40 | 0.8238 | 0.8223 | 0.8168 | **0.8319** | 0.7737 | 0.8230 | 0.8298 | 0.8312 |
| 50 | 0.8148 | 0.8195 | 0.8262 | 0.8267 | 0.8049 | 0.8287 | 0.8346 | **0.8425** |
| 60 | 0.8148 | 0.8360 | 0.8350 | 0.8227 | 0.8220 | 0.8304 | 0.8358 | **0.8388** |
| 70 | 0.8099 | 0.8195 | 0.8268 | 0.8209 | 0.8125 | 0.8321 | **0.8385** | 0.8368 |
| 80 | 0.8139 | 0.8275 | 0.8298 | 0.8304 | 0.8305 | 0.8294 | 0.8311 | **0.8357** |
| 90 | 0.8122 | 0.8409 | 0.8357 | 0.8251 | 0.8170 | 0.8311 | 0.8390 | **0.8452** |
| 100 | 0.8176 | 0.8176 | 0.8176 | 0.8176 | 0.8176 | 0.8176 | 0.8176 | 0.8176 |

Table 1: Tabulated F1 results for SST2 dataset with added baselines of $k$-Means and DeDup .

| % | Random | AlpaGasus | Forgetting | Prototypicality | EL2N | $k$-Means | SemDeDup | ACS |
|---|--------|-----------|------------|-----------------|------|-----------|----------|-----|
| 10 | 0.2293 | 0.2393 | 0.2427 | 0.2508 | 0.2597 | 0.2314 | 0.2506 | **0.2642** |
| 20 | 0.3000 | 0.3235 | 0.3157 | 0.3102 | 0.3191 | 0.3164 | 0.3162 | **0.3352** |
| 30 | 0.3486 | 0.3512 | 0.3444 | 0.3487 | 0.3391 | 0.3583 | 0.3609 | **0.3732** |
| 40 | 0.3494 | 0.3568 | 0.3566 | 0.3494 | 0.3579 | 0.3598 | 0.3724 | **0.3744** |
| 50 | 0.3595 | 0.3520 | 0.3677 | 0.3489 | 0.3619 | 0.3601 | 0.3687 | **0.3730** |
| 60 | 0.3546 | 0.3647 | 0.3597 | 0.3609 | 0.3648 | 0.3652 | 0.3598 | **0.3684** |
| 70 | 0.3634 | 0.3542 | 0.3682 | 0.3665 | 0.3640 | 0.3689 | 0.3690 | **0.3749** |
| 80 | 0.3512 | 0.3618 | 0.3695 | 0.3647 | **0.3663** | 0.3555 | 0.3644 | 0.3654 |
| 90 | 0.3689 | 0.3681 | 0.3604 | 0.3628 | 0.3676 | 0.3724 | 0.3749 | **0.3785** |
| 100 | 0.3729 | 0.3729 | 0.3729 | 0.3729 | 0.3729 | 0.3729 | 0.3729 | 0.3729 |

Table 2: Tabulated F1 results for FewRel dataset with added baselines of $k$-Means and SemDeDup .

## C    REBUTTAL EXPERIMENTS

### C.1    ADDITIONAL BASELINES

We here provide experimental results for the suggested additional baselines from the reviewers. Specifically, we implement both $k$-Means and the semantic de-duplication method of (Abbas et al., 2023).

For the implementation of $k$-Means, we adapt off-the-shelf $k$-Means clustering from SciPy on the data Gecko embeddings, and set $k$ equal to the desired number of selections. For SemDeDup, we first cluster using $k$-Means to select $k/10$ cluster centers and subsequently sample points according to the SemDeDup similarity based pruning of the dataset until we obtain the desired number of samples.

Throughout the experiments, we observe that $k$-Means does worse than ACS, and SemDeDup achieves performance roughly between the $k$-Means and ACS. This is consistent with expectation since the method is methodologically a midpoint between the two methods.

| % | Random | AlpaGasus | Forgetting | Prototypicality | EL2N | k-Means | SemDeDup | ACS |
|---|---|---|---|---|---|---|---|---|
| 10 | 0.1820 | 0.1850 | 0.1905 | 0.1789 | 0.1917 | 0.2031 | 0.2456 | **0.2502** |
| 20 | 0.3122 | 0.3283 | 0.3142 | 0.3394 | 0.3202 | 0.3409 | 0.3575 | **0.3851** |
| 30 | 0.3289 | 0.3459 | 0.3366 | 0.3523 | 0.3526 | 0.3582 | 0.3648 | **0.3830** |
| 40 | 0.3483 | 0.3547 | 0.3434 | 0.3696 | 0.3674 | 0.3684 | 0.3700 | **0.3806** |
| 50 | 0.3675 | 0.3593 | 0.3538 | 0.3747 | 0.3699 | 0.3611 | 0.3774 | **0.3820** |
| 60 | 0.3745 | 0.3764 | 0.3721 | 0.3736 | 0.3867 | 0.3839 | 0.3798 | **0.3921** |
| 70 | 0.3673 | 0.3788 | 0.3745 | 0.3727 | 0.3860 | 0.3803 | 0.3923 | **0.3945** |
| 80 | 0.3731 | 0.3814 | 0.3785 | 0.3763 | 0.3812 | 0.3852 | 0.3859 | **0.3958** |
| 90 | 0.3745 | 0.3787 | 0.3809 | 0.3781 | 0.3788 | 0.3851 | 0.3840 | **0.3877** |
| 100 | 0.3842 | 0.3842 | 0.3842 | 0.3842 | 0.3842 | 0.3842 | 0.3842 | 0.3842 |

Table 3: Tabulated F1 results for CrossNER dataset with added baselines of $k$-Means and DeDup .

## C.2 LABEL SELECTION BREAKDOWN

To rigorously assess whether the implemented unsupervised downsampling methods maintain the original distribution of data labels, we compute the Total Variation Distance (TVD) between the empirical label distribution of the selected subsets and the uniform distribution. Given that the source synthetic datasets are generated to be perfectly balanced across $N$ classes, a balanced subset selection should maintain a uniform prior $U$ where $U(c) = 1/N$ for all classes $c$. Let $P_k$ be the empirical probability distribution of labels within a selected subset of size $k$. We quantify the divergence from uniformity using the TVD, defined as half the $L_1$ distance between the distributions:

$$\delta(P_k, U) = \frac{1}{2} \sum_{c=1}^{N} |P_k(c) - U(c)|$$

The value $\delta(P_k, U) \in [0, 1 - 1/N]$ serves as a measure of displaced probability mass. A TVD near 0 indicates that the selection method preserves the original class balance (representational fairness), while a high TVD indicates that the method over selects certain classes for training. This analysis is particularly interesting for ACS, as the method operates without access to label supervision; a low TVD empirically validates that maximizing graph coverage serves as an effective proxy for label diversity.

We exclude the CrossNER dataset from this specific analysis due to a structural mismatch between the selection unit and the labeling unit. In CrossNER, ACS selects distinct sentences based on their embedding similarity, but the downstream task requires classifying individual tokens (Named Entity Recognition). Since a single sentence may contain an arbitrary number of entity tokens across various classes (or none), the resulting distribution of token-level labels is a second-order effect of sentence selection. Consequently, comparing the token-label distribution to a sentence-level uniform prior is ill-defined and does not accurately reflect the semantic coverage of the selection algorithm.

| % | AlpaGasus | Forgetting | Prototypicality | EL2N | ACS |
|---|---|---|---|---|---|
| 10 | 0.0200 | 0.0200 | 0.1000 | 0.0200 | 0.1600 |
| 20 | 0.0275 | 0.0261 | 0.0087 | 0.0058 | 0.0652 |
| 30 | 0.0039 | 0.0039 | 0.0047 | 0.0078 | 0.0516 |
| 40 | 0.0021 | 0.0027 | 0.0144 | 0.0123 | 0.0422 |
| 50 | 0.0020 | 0.0020 | 0.0203 | 0.0118 | 0.0455 |
| 60 | 0.0003 | 0.0003 | 0.0131 | 0.0052 | 0.0567 |
| 70 | 0.0030 | 0.0030 | 0.0066 | 0.0038 | 0.0588 |
| 80 | 0.0026 | 0.0026 | 0.0069 | 0.0021 | 0.0527 |
| 90 | 0.0031 | 0.0031 | 0.0027 | 0.0052 | 0.0477 |

Table 4: SST2 Total Variation Distance computer for each subset selection method and fraction of data selected.

| % | AlpaGasus | Forgetting | Prototypicality | EL2N | ACS |
|---|---|---|---|---|---|
| 10 | 0.0991 | 0.0837 | 0.0960 | 0.0806 | 0.0907 |
| 20 | 0.0770 | 0.0654 | 0.0717 | 0.0630 | 0.0670 |
| 30 | 0.0641 | 0.0586 | 0.0585 | 0.0610 | 0.0588 |
| 40 | 0.0588 | 0.0508 | 0.0525 | 0.0550 | 0.0523 |
| 50 | 0.0528 | 0.0466 | 0.0496 | 0.0501 | 0.0470 |
| 60 | 0.0468 | 0.0402 | 0.0425 | 0.0478 | 0.0441 |
| 70 | 0.0424 | 0.0378 | 0.0407 | 0.0473 | 0.0398 |
| 80 | 0.0400 | 0.0388 | 0.0419 | 0.0466 | 0.0399 |
| 90 | 0.0400 | 0.0389 | 0.0395 | 0.0432 | 0.0396 |

Table 5: FewRel Total Variation Distance computer for each subset selection method and fraction of data selected.

