# OpenReview forum: "Less is More: Adaptive Coverage Sampling for Synthetic Training Data"
_ICLR.cc/2026/Conference — Submitted to ICLR 2026_

### Official Review · Reviewer_Rg75 · 2025-10-27

**Soundness:** 3
**Presentation:** 2
**Contribution:** 4
**Rating:** 6
**Confidence:** 4

**Summary:**

This paper proposes Adaptive Coverage Sampling (ACS), a novel and theoretically grounded reformulation of the data selection problem as a maximum coverage problem on a similarity graph. The key idea is to represent synthetic data samples as nodes connected by semantic similarity edges and to identify a compact subset of samples that maximizes representational coverage. The authors establish a monotonicity theorem showing that coverage increases as the similarity threshold decreases, enabling an efficient binary search to find the optimal threshold. They further demonstrate empirically that this monotonicity holds in practice even under greedy approximation. Extensive experiments across multiple NLP tasks show that ACS consistently outperforms heuristic and LLM-based baselines, achieving comparable or superior performance using only a small fraction of the synthetic data. Interestingly, the results also suggest that smaller, carefully selected subsets can sometimes improve model performance compared to training on the entire synthetic dataset, highlighting the effectiveness of principled data reduction.

**Strengths:**

* The paper presents a novel and theoretically grounded formulation of the data selection problem as a graph-based maximum coverage optimisation. This framing is both elegant and insightful, connecting efficient data selection with submodular optimisation and thereby enabling the use of greedy algorithms with established approximation guarantees.

* The authors provide a sound theoretical analysis for the ideal (exact) case and an empirical validation showing that the monotonicity property also holds under the greedy approximation. This property justifies the use of an efficient binary search to automatically determine the optimal similarity threshold for graph construction.

* The paper addresses a timely and practically important challenge: how to effectively sample and curate synthetic data to improve downstream model performance, a problem that is becoming increasingly relevant as synthetic data becomes more widespread.

* The paper includes comprehensive experimental evaluations across multiple NLP tasks, demonstrating that ACS achieves significant efficiency gains over existing methods.

**Weaknesses:**

* Although the formulation of the paper is compelling, the writing sometimes makes it difficult to follow the motivation behind searching for the maximum threshold that achieves sufficient coverage. In particular, the connection between threshold selection and downstream performance is not always clearly motivated or illustrated.

* The proposed ACS method implicitly assumes that the synthetic-data distribution and the target downstream distribution are closely aligned, meaning that nearly all relevant points are coverable in the constructed graph. However, in synthetic-data settings (as the paper emphasises), this assumption may not hold: synthetic data can include hallucinations, label inconsistencies, or missing regions of the target space. Recent works [1], [2] argue that distribution alignment and data quality are as important as diversity or coverage. The paper would be strengthened by explicitly discussing this assumption and clarifying under what conditions ACS remains effective when the synthetic and target distributions diverge.

* The theoretical connection between the maximum-coverage solution (based on the maximum threshold) and other established objectives, such as maximising information gain under identical training and testing distributions, is not explored. Establishing such a connection could strengthen the theoretical impact and situate the contribution more firmly within the broader literature.

* The computation of pairwise similarities can be costly, particularly for large datasets. In addition, obtaining high-quality embeddings often requires using larger and more expressive models, which independently increases the computational and memory cost of the overall method.

[1] Kuo et al., Not All LLM-Generated Data Are Equal: Rethinking Data Weighting in Text Classification, ICLR 2025

[2] Li et al., Synthetic Data Generation with Large Language Models for Text Classification: Potential and Limitations, EMNLP 2023

**Questions:**

* The citation formatting could be improved. Many in-text citations use `\citet`, while `\citep` would be more appropriate for parenthetical references in formal academic writing.

* The ACS method builds on the submodular property of the coverage function, but the similarity threshold is determined through binary search rather than being incorporated directly into the objective. Could the authors explain why they did not consider alternative monotone submodular objectives, such as information gain [3] or a weighted coverage formulation [3], defined as
  $$f(S) = \sum_{(i,j)\in E} s(i,j) \mathbf{1}(i \in S \text{ or } j \in S),$$
  where $s(i,j)$ denotes the similarity between nodes $i$ and $j$, $E$ is the set of edges, and $\mathbf{1}(\cdot)$ is an indicator function that counts each edge once if either endpoint is selected. This weighted formulation remains monotone and submodular, retains the same $(1 - 1/e)$ approximation guarantee of the greedy algorithm, and removes the need to tune a similarity threshold.


* The ACS framework depends on two key hyperparameters: the coverage target $c$ and the budget $k$. How sensitive are the results to these parameters, and how should they be selected in practice?

[3] Andreas Krause and Daniel Golovin. Submodular Function Maximization. In Tractability: Practical Approaches to Hard Problems, Cambridge University Press, 2014.

---

> ### Author Response · Authors · 2025-12-04
> **Response to ReviewerRg75**
>
> We appreciate the reviewer’s thorough engagement with our work and for highlighting the "elegant and insightful" nature of our graph-theoretic formulation! We value the high contribution score and the questions regarding alternative submodular objectives. We have addressed the noted presentation issues and here address these methodological and theoretical points below.
>
> *``...why not consider alternative submodular objectives...''*
>
> While other weighted coverage (like facility location) is a natural objective to optimize for, we chose our hard thresholding with binary search for the the following reasons. First, in the semantic embedding space utility cannot be considered linear. Concretely, a weighted sum approach might consider 9 weak neighbors (similarity of 0.1) and one strong neighbor (similarity 0.9) equivalent. The hard thresholding approach instead ensures that coverage is only counted for samples which are more likley to be semantically interchangeable with the covered sample. Second, distributions of similarity change between datasets (notably, SST2 is dense compare to the other tasks). Thus, a weighted objective would need to be further tuned per dataset, whereas our approach can independently learn the requisite threshold from the data.
>
> *``...synthetic and downstream data distributions...''*
>
> We agree that distributional mismatch is a challenge, but clarify that ACS is implicitly robust to this as a result of the max-coverage objective.
> Specifically, hallucinations and label noise will appear as outliers in the embedding space when compared against the more common, grounded, samples.
> By prioritizing high-degree nodes (dense regions of the graph) outlier hallucinations are unlikely to be sampled.
> We will expand on this point with reference to your noted citation (Kuo et al and Li et al) to better contextualize these ideas. Thank you for the relevant works.
>
> *``...connection between threshold selection and downstream performance...''*
>
> We clarify that the threshold defines a semantic resolution for the dataset (and thus, the graph topology). If this threshold is too high, the graph becomes disconnected and the algorithm acts more like random sampling. If the threshold is too low, the graph moves toward larger connected components and the algorithms selects only a few central points to cover the whole dataset. Optimizing this value finds the critical point where the desired subset size, $k$, is just large enough to cover the dataset at the highest possible resolution.
>
> *``Computation of pairwise similarities can be costly...''*
>
> We again highlight Appendix B, where we proved and empirically validated that the optimal threshold can actually be estimated on a small random subsample of the dataset with bounded error. This avoids the full $O(N^2)$ computation. Additionally, there exists an extensive literature on scalable graph building via methods such as Locality Sensitive Hashing [1,2].
>
> [1] "Scalable Graph Representation Learning via Locality Sensitive Hashing" Chen et al. 2022
>
> [2] "Data Sampling Using Locality Sensitivity Hashing for Large Scale Graph Learning" Shekkizhar. 2023

---

### Official Review · Reviewer_GnUP · 2025-10-30

**Soundness:** 2
**Presentation:** 2
**Contribution:** 2
**Rating:** 2
**Confidence:** 4

**Summary:**

This work proposes selecting a representative subset from the generated dataset by framing it as a maximum-coverage problem. The algorithm allows using only 10% of the data to achieve similar performance.

**Strengths:**

1. The algorithm is intuitive.
2. The experiments demonstrate improvement over several baselines, including random selection and AlphaGasus.

**Weaknesses:**

1. The main experiment is based on data generated by GPT-3.5. I am doubtful we can achieve similar results with more recent models like GPT-4o-mini, which should be much cheaper and more intelligent.
2. While the description of the used baseline is limited, I can see that a lot of data selection baselines are missing: (I) k-center / k-medoids / coverage and other classic algorithms. (I) Deduplication approach used in pretraining [1][2] and finetuning [3]

[1] D4: Improving LLM Pretraining via Document De-Duplication and Diversification

[2] SemDeDup: Data-efficient learning at web-scale through semantic deduplication

[3] NLU on DataDiets: Dynamic Data Subset Selection for NLP Classification Tasks

**Questions:**

See Weaknesses

---

> ### Author Response · Authors · 2025-12-04
> **Response to Reviewer GnUP**
>
> We thank the reviewer for their assessment and for finding our algorithm intuitive. We address the two specific concerns regarding generator models and baselines below.
>
> *``Generated by GPT3.5...''*
>
> We respectfully believe that the utility of ACS is agnostic to the generator model.
> Furthermore, we here use publicly available synthetic datasets from prior works so as to control for any potential prompt engineering.
> Thus, our experiments control for confounding variables and isolate the impact of the selection methods exclusively.
> While we agree that newer models may generate better data, we do not believe that this undermines the utility and improvements possible with the ACS procedure in selecting diverse data subsets.
>
> *``Missing baselines...''*
>
> As noted in our response to Reviewer 8cs9, the greedy approach is the known optimal approximation algorithm for monotone submodular maximization hence the selection of this heuristic.
> We further note that $k$-centers objective is more sensitive to outliers, which is a common issue with sensitive data that we here work to overcome. As such, max coverage is more robust in capturing the core data points from a distribution.
> Nonetheless, we discuss experimental results for additional baseline selection methods in the general comment to all reviewers and Appendix C of the revision.

---

### Official Review · Reviewer_8cs9 · 2025-10-31

**Soundness:** 3
**Presentation:** 3
**Contribution:** 2
**Rating:** 6
**Confidence:** 4

**Summary:**

This paper formulates downsampling and de-redundancy as a graph-based maximum k-coverage problem and uses a greedy approximation with threshold binary search; experiments on SST-2, FewRel, and CrossNER-AI validate its advantages.

**Strengths:**

1. This paper is the first to model de-redundancy of LLM-synthesized datasets as a graph based maximum k coverage problem; it provides a formal definition of coverage and proves that, under a similarity thresholded graph, the optimal maximum coverage solution is monotone in the threshold.
2. Across SST-2, FewRel, and CrossNER-AI, the method often surpasses baselines
3. Scalability is handled by tuning the threshold on a small subsample and transferring it to the full set; a Hoeffding-type bound supports this, and thresholds learned from under 20% of the data reliably meet the target coverage, with or without a degree cap.

**Weaknesses:**

1. When the de-redundancy of LLM-synthesized datasets is reduced to a graph-based maximum k-coverage problem, the employed approach using **greedy approximation** and **threshold binary search** is standard practice rather than an innovation. For example:
[1] Lin, Hui, Jeff Bilmes, and Shasha Xie. "Graph-based submodular selection for extractive summarization." *2009 IEEE Workshop on Automatic Speech Recognition & Understanding*. IEEE, 2009.
[2] Biabani, Leyla, et al. "Faster Query Times for Fully Dynamic $ k $-Center Clustering with Outliers." *Advances in Neural Information Processing Systems* 36 (2023): 9226-9247.
2. The experimental evaluation remains insufficient.
    - It does not compare against alternative algorithms for the maximum k-coverage problem, including submodular maximization and k-center-based solvers.
    - It omits head-to-head comparisons with coverage-centric coreset methods that target representativeness under a fixed budget.
    - It lacks baselines from standard dataset de-duplication pipelines, such as near-duplicate detection, leaving the advantages over de-redundancy untested.

**Questions:**

See weakness above.

---

> ### Author Response · Authors · 2025-12-04
> **Response to Reviewer 8cs9**
>
> We thank the reviewer for their thoughtful evaluation and for recognizing the strengths of our work, particularly the formal definition of coverage, the proof of monotonicity , and the rigorous handling of scalability via the Hoeffding-bound transferability. We address the concerns regarding novelty and baselines below.
>
> *``...the employed approach uses greedy approximation and threshold binary search is standard...''*
>
> We thank the reviewer for the noted references (to be cited in the final version of our paper).
> We agree that the greedy algorithm and binary search are not novel.
> However, we respectfully clarify that our contribution is the novel constraint satisfaction approach employed here, specifically for synthetic data distillation.
> Unlike the approach of Lin et al., ACS introduces a max-degree constraint and the problem of inverse solving for the coverage target. Both of which considerably alter the optimization problem.
> By constraining the degree of nodes, we force the process to ignore generic hubs and focus on selecting a larger, more diverse and locally representative, set of samples.
> Furthermore, we utilize the monotonicity of the coverage function to solve for the threshold $\tau$ which guarantees a specific target level. While the algorithmic methods are known, the actual implementation in the present context is non-trivial.
>
> *``...does not compare against alternative coverage algorithms''*
>
> In our problem, the greedy approach is the known optimal approximation algorithm for monotone submodular maximization hence the selection of this heuristic.
> We further note that $k$-centers objective is more sensitive to outliers, which is a common issue with sensitive data that we here work to overcome. As such, max coverage is more robust in capturing the core data points from a distribution.
> Nonetheless, we discuss experimental results for additional baseline selection methods in the general comment to all reviewers and Appendix C of our revised pdf.
>
> *``...de-duplication baselines''*
>
> As noted in our general comment to all reviewers, we have implemented the SemDeDup and $k$-Means baselines in Appendix C of the revised submission.

---

### Official Review · Reviewer_GKGn · 2025-11-03

**Soundness:** 2
**Presentation:** 1
**Contribution:** 2
**Rating:** 2
**Confidence:** 3

**Summary:**

This paper proposes a method to select representative subsets (subsets) of data points from a synthetically generated corpus. It relies in building a similarity graph over the data points, i.e., data points are nodes and edges similarities. The with a greedy graph traversal a subset of nodes is extracted. The hyper-parameters are the threshold on edge similarities and number of extracted nodes. Varying these two it is possible to obtain different samples with different diversity and coverage of the underlying entire corpus. The motivation for such proposal is the usually observed data imbalance and bias amplification in synthetic datasets.

**Strengths:**

- A method to select varied samples for data imbalance.

**Weaknesses:**

- Weak motivation. That LLMs serve to generate training data for downstream tasks and then fine-tune small models. If the LLM model can generate data, then can be used directly to solve the task.

- Differences between methods seem small (Figure 3/4/5). Bigger differences seem in regions where F1 scores are not the best.

- Experiments are carried out on old BERT models. It is unclear how the sampling method would help newer LLMs.

- Scalability of the approach. A method for efficient and high quality data sampling would be very useful for very large corpora (i.e., LLM pre-training and fine-tuning). The graph similarity construction seems to be rather expensive in these cases.

**Questions:**

- Sample variation is assessed via SelfBLEU, is it possible to also show number of instances per class in the target samples (in comparison also with the exiting reported methods).

---

> ### Author Response · Authors · 2025-12-04
> **Response to Reviewer GKGn**
>
> We thank the reviewer for their comments on our work. We here address the noted weaknesses and questions.
>
> *``If the LLM model can generate the data, then can be used to directly solve the task.''*
>
> We respectfully disagree with this premise.
> This view overlooks several fundamental paradigms in NLP research and engineering.
> Specifically, synthetic data is critical in the domains of knowledge distillation, red-teaming, self-improvement, and evaluation. We further highlight that mitigating redundancy and identifying diverse subsets of data in the NLP domain was noted as a critical problem in the field by [1].
>
>
> *``Differences seem small...''*
>
> We emphasize that the primary contribution of our work lies in data efficiency, not in training the best possible downstream models.
> Crucially, our results show that ACS achieves comparable performance for downstream models by selecting only 10-30\% of the data. This represents a massive reduction in the computational overhead for fine-tuning.
>
> *``Experiments on old BERT models...''*
>
> We recognize that BERT is an older model, but note that this was by design to carefully isolate the data selection variable from the model capability variable, following standard data pruning literature.
> We refer the reviewer to our general comment for further discussion on this point.
>
> *``Scalability of the approach...''*
>
> We respectfully point to Appendix B for more detailed comments on the scalability of ACS.
> Specifically, in the Appendix we show that we do not need to construct the full similarity graph to identify the optimal similarity threshold--a small subset of the data suffices to approximate this value (we demonstrate this both in theory and practice).
> Furthermore, there exists an extensive literature on scalable graph building via methods such as Locality Sensitive Hashing [2,3].
>
> *``Is it possible to also show number of instances per class...''*
>
> In this revised version of our paper we provide a complete analysis of the label selections by ACS as compared to the baseline methods by computing the Total Variational Distance between the selection distributions and the original data.
> We refer the reviewer to Appendix C for these results.
>
> [1] "Best practices and lessons learned on synthetic data." Liu et al. 2024
>
> [2] "Scalable Graph Representation Learning via Locality Sensitive Hashing" Chen et al. 2022
>
> [3] "Data Sampling Using Locality Sensitivity Hashing for Large Scale Graph Learning" Shekkizhar. 2023

---

### Author Response · Authors · 2025-12-04
**General Comment to Reviewers**

We thank the reviewers for their thoughtful feedback and engagement with our submission. We below address individual concerns, and here discuss the suggested additional experiments which have been conducted for the revised version of our manuscript.

1. Comparison against $k$-means and deduplication: In the appendix of our revised manuscript, we provide results comparing ACS with $k$-means and SemDeDup [1] for selecting representative samples. We see a similar trend to the results presented in our submission: ACS consistently selects more informative samples which leads to improved downstream performance.

2. Fine-tuning newer models:  We recognize that BERT is an older model, but note that this was initially by design to carefully isolate the *data selection* variable from the *model capability* variable. Moreover, utilizing modern LLMs as the downstream classifier introduces the confounding factor of a model's extensive pre-trained world knowledge.
BERT here serves as a sensitivity probe: if ACS allows a BERT model to match full-dataset performance with only 20\% of sample, this demonstrates that our method is well-equipped to identify the core semantic subsets of a larger corpus.
Furthermore, we note that ACS operates purely on the geometry of a selected embedding space (here Gecko) and is agnostic to the downstream model. Thus, improvements on the selection transfer to larger architectures.
We reiterate that, in the present work, we are measuring the data selected, rather than downstream task performance (but the latter serves as a proxy for the former).

[1] "SemDeDup: Data-efficient learning at web-scale through semantic deduplication" Abbas et al 2023

---

### Meta-Review · Area_Chair_pp2v · 2026-01-07

**Summary:**

The paper presents a novel approach to selecting representative subsets from synthetic datasets using Adaptive Coverage Sampling (ACS), addressing redundancy and imbalance. Reviewers questioned the fundamental need for the proposed data selection method, suggesting that if synthetic data can be generated by LLMs, it should directly serve the task. The novelty of the proposed method was also questioned, with reviewers noting that the algorithmic techniques used, such as greedy approximation and binary search, are not new. Additionally, concerns about the scalability of the approach, especially regarding the computational cost of pairwise similarity calculations for large datasets, were only partially addressed. Despite some empirical results, the lack of significant novelty in algorithmic contribution and the practical challenges in scaling the method to large datasets indicate that this work does not offer a sufficiently compelling contribution to justify acceptance.

**Reviewer Concerns:**

Concerns addressed
- Reviewer GKGn questioned the necessity of selecting a subset of synthetic data, the authors clarified that synthetic data generation is crucial for knowledge distillation, red-teaming, and other NLP paradigms.
- Reviewer GnUP raised concerns that the core graph-theoretic max-cover formulation was not sufficiently engaged with and recommended considering alternative submodular objectives. The rebuttal addressed this concern by highlighting the uniqueness of the problem formulation and the specific constraints applied.

Concerns Still Outstanding:
- Reviewer 8cs9 noted that the greedy approximation and binary search techniques used are standard and questioned the novelty of the approach. While the authors provided justification for their choice of algorithm, the fundamental concern about the lack of novel algorithmic contribution still stands, as these methods are widely used in optimization problems without significant new theoretical contributions.
- Reviewer GKGn and Reviewer Rg75 questioned the scalability of the method, particularly regarding the computational cost of pairwise similarity calculations for large datasets. While the authors addressed the scalability concern by explaining how to approximate the threshold with reduced computational cost, the reviewers' concerns about the method’s scalability to extremely large datasets have not been fully alleviated. The use of LSH was mentioned but not sufficiently demonstrated or validated in the experiments.

**Reviewer Scores:**

There is no clear evidence that any reviewer would change his/her score.

---

### Decision · Program_Chairs · 2026-01-26

Reject